# Referring Expression Matters: Multi-referring Feature Aggregation for Referring Video Object Segmentation

## Abstract

Referring Video Object Segmentation aims to segment object instances referred to by natural language referring expressions in a video sequence. This interaction style is quite simple and flexible, being capable of producing high quality segmentation masks. However, the referring expression variation occurs due to the randomness of expressions provided by users, making the existing state-of-the-art models still face the problem of wrongly identifying the referred object. To address this issue, we present a novel referring video object segmentation network fed with multiple referring expressions. Specifically, a simple but effective neural expression generation module is proposed to map the features of multiple referring expressions to complementary features with less redundancy. This interaction of multiple referring expressions not only is beneficial to identify the referred object but also speeds up the training convergence. We make evaluations of the proposed method on the popular referring video object segmentation datasets, and experimental results demonstrate that our method outperforms the state-of-the-arts by a significant margin in terms of segmentation quality and achieves considerable gains in terms of training convergence speed. Our code and pre-trained models will be available.

## 1 Introduction

Video object segmentation is a fundamental task in computer vision, aiming at the segmentation of object instances across a video sequence. To specify the object to be segmented, various types of user input have been explored up to now, including unsupervised (no manual annotation), semi-supervised (1st frame annotation) and interactive (scribble or click annotation) types, etc. Considering that the language is often used in our daily life to indicate a specific object, a new task, called Referring Video Object Segmentation (RVOS), is introduced by the methods (Gavrilyuk et al., 2018) and (Khoreva et al., 2018). The RVOS task exploits language to identify and segment an object referred to by the given language description (a.k.a. referring expression RE), playing an important role in human-computer interaction, robotics, and video editing applications.

To accurately segment the referred object in a video clip, one needs to holistically understand the RE and video content. When the model fails to understand the input RE, this might lead to the completely wrong mask segmentation. As a fact, the significance of linguistic structure in the RE is found in syntactic and lexical ablation experiments that were conducted on the RE segmentation datasets (Cirik et al., 2018; Bellver et al., 2023), where both the length of the RE and combinations of semantic attributes are reported making a difference to the segmentation performance.

To figure out why this happens, understanding the generation of the RE seems somewhat helpful. Now, the approaches of generating the RE in the existing referring expression segmentation datasets could be mainly grouped into 1) template-based, and 2) free-form-based. The former typically generates the expressions by adding categories, attributes, and relationships (Wu et al., 2020), while the latter generates natural expressions by Amazon Mechanical Turk workers that are asked to accurately describe the target object (Seo et al., 2020). In contrast, the free-form is mostly used, since it is simpler and less constrained. Due to the randomness of expressions provided by users, the RE

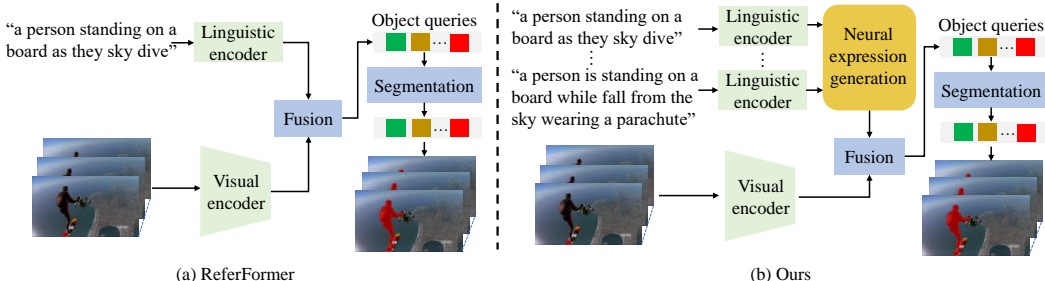

Figure 1: Conceptual comparison between (a) the top-performing method ReferFormer (Wu et al., 2022b) and (b) our proposed method.

variation occurs, which might explain the above significant influence of the RE on the segmentation performance.

To address the above-mentioned issue, we return to the user guidance: the RE. In a real scenario, we could imagine that when talking with a person or a voice assistant (e.g. "Siri"), if one sentence is not effective enough, we typically change the RE and provide a second one with similar meanings to enhance the understanding. This interaction raises one question: *will it improve the accuracy of the RVOS model with the input of multiple referring expressions?*

W.r.t. multiple referring expressions (MRE), as shown in Table 1, the large-scale RVOS dataset Ref-Youtube-VOS (Seo et al., 2020) provides almost two referring expressions per target instance. The A2Dre+ dataset (Bellver et al., 2023) generates many more referring expressions for each target instance through referring expression categorization, enabling the analysis of how referring expressions influence RVOS. Besides, earlier RVOS datasets Ref-DAVIS16 and Ref-DAVIS17 (Khoreva et al., 2018) offer two more referring expressions per instance. Actually, MRE for the same object instance are used by the existing RVOS models in the training, during which the models are trained with one RE at each iteration. However, they are still limited in predicting the correct object, which might put down to no interaction between MRE. Nevertheless, the combination of semantic attributes through the fashion of feeding MRE to the network is still unexplored, which is the research focus of this paper.

Apart from the RE understandings, an efficient end-to-end framework precisely segmenting objects of interest is also indispensable. With recent advances in object detection and segmentation, DEtection TRansformer (DETR) (Carion et al., 2020) has emerged as a powerful framework that yields state-of-the-art quality. Top-performing methods for the RVOS are based on the DETR-like framework. Despite significant improvements that have been achieved, DETR-based methods still suffer from slow training convergence. For each referred object given with $M^1$ number of referring expressions, if the input number of referring expressions of the network is $M^2$, the total number of iterations at each epoch will be decreased from $M^1$ to $\frac{M^1}{M^2}$. Compared with a single RE used per iteration, the interaction of $M^2$ referring expressions has more potential to generate effective semantic referring features. Here raises another question: *is it possible to improve the speed of training convergence of the DETR-based model for RVOS with well-aggregated MRE?*

In this paper, we make an attempt to seek the answer to two raised questions. Our proposed method in Figure 1 (b) is built on top of the top-performing method ReferFormer (Wu et al., 2022b) in Figure 1 (a). To tackle the RE variation, we propose a simple but effective module Neural Expression Generation (NEG) to learn a mapping from multiple referring expression features to the more representative features. In addition, we propose optimization improvements such as sampling more points around the referred object to aggregate more context in the fine-tuning phase, so as to improve the ability to handle objects with varying shapes. To summarize, we make the following contributions:

- The first RVOS method that is fed with multiple referring expressions to obtain the complete and concise linguistic features by a simple yet effective neural expression generation module.

Table 1: Summary of referring video object segmentation datasets. # RE: the number of Referring expressions.

| Dataset | Videos | #RE per Instance | Form |
|---|---|---|---|
| Ref-DAVIS16 | 50 | 4 | Free-form |
| Ref-DAVIS17 | 90 | 4 | Free-form |
| Ref-Youtube-VOS | 3975 | $\approx 2$ | Free-form |
| A2Dre+ | 190 | $\approx 8$ | - |

- Proposed different sampling strategies in cross-modal attention for pre-training and fine-tuning to boost the model performance.
- Having achieved the superior accuracy over state-of-the-art methods, and $2\times$ faster training convergence compared to the top-performing method ReferFormer.

## 2 RELATED WORK

### 2.1 REFERRING VIDEO OBJECT SEGMENTATION

The objective of RVOS is segmenting an object across a video sequence with given referring expressions. Some early works made an attempt to tackle this task in a two-stage fashion, i.e. decomposing it into well-known sub-sequential tasks. Khoreva *et al*. (Khoreva et al., 2018) present a "localize-then-segment pipeline", where the referring expression is used to localize the target object, and then pixel-wise segmentation network is employed to obtain the mask. Seo *et al*. (Seo et al., 2020) present two sequential neural networks (URVOS) and semi-supervised video object segmentation respectively, in which one mask with a high confidence is predicted and then propagated to other frames. Recently, the one-stage learning-based method stood out to be a prevalent method for the RVOS, which could be classified into two-stream and single-stream spatial-temporal interaction methods. Ding *et al*. (Ding et al., 2022) propose a two-stream model architecture (LBDT) that takes into account the appearance and frame differences, in which linguistic features are leveraged to bridge spatial-temporal interactions. Zhao *et al*. (Zhao et al., 2022) propose a multi-modal alignment loss to align appearance, optical flow, and linguistic features. Regarding single-stream solutions, Gavrilyuk *et al*. (Gavrilyuk et al., 2018) propose to encode linguistic features as kernels of convolutional dynamic filters to convolve visual features. Li *et al*. (Li et al., 2022) present a meta-transfer approach (YOFO) to disregard irrelevant linguistic features and transfer useful linguistic features to the outputs that are dominated by spatial-temporal features. Wu *et al*. (Wu et al., 2022a) propose to model multi-modal interactions by dynamic semantic alignment in a multi-level manner: video-level, frame-level, and object-level. Liang *et al*. (Liang et al., 2023) present a computation-efficient and memory-efficient fully attentional Transformer framework (LOCATER). Hui *et al*. (Hui et al., 2023) present a language-aware spatial-temporal collaboration framework (CSTM), in which language features are updated progressively. More recently, query-based DETR-like models (Adam et al., 2022; Wu et al., 2022b) have emerged as a powerful alternative to previous one-stage methods. Multi-modality interaction is simply implemented by the concatenation of linguistic and vision features or early cross-attention fusion of linguistic and vision features (Wu et al., 2022b), followed by the transformer encoder. The temporal coherency is achieved by linking the corresponding indexes across video object queries. Though these methods have achieved impressive results in some scenarios, but their performance is still limited by the understanding of referring expressions (for instance, cf. Figure 3).

### 2.2 REFERRING EXPRESSION GENERATION

To generate referring expressions for natural scenes, previous works mainly follow two directions: template-based and free-form-based.

**Template-based.** Kazemzadeh *et al*. (Kazemzadeh et al., 2014) present an automatic method to generate template-based referring expressions from sentences in image retrieval datasets under a set of pre-defined attributes, e.g. category, color, relative location, etc. Wu *et al*. (Wu et al., 2020) generate referring expression annotations by adding categories, attributes, and relationships, based

on the Visual Genome dataset which includes images and scene-graph annotations.

**Free-form-based.** Kazemzadeh *et al*. (Kazemzadeh et al., 2014) present a free-form-based method, in which a two-player game is developed, and posted online for players to generate expressions referring to objects and click on the location of referred objects. To generate Ref-DAVIS17 (Khoreva et al., 2018) and Ref-Youtube-VOS dataset (Seo et al., 2020), annotators were given a pair of videos, i.e. the original video and the mask-overlaid video with the target object highlighted, and then asked to describe the target object accurately. The natural language expressions in the RVOS dataset behave differently on aspects of the length, semantic attributes, and so on. The interaction between natural language expressions for the same object instance has not been explored in existing RVOS models which are usually trained with one referring expression at each iteration.

### 2.3 TRANSFORMERS

The Transformer (Vaswani et al., 2017) has been introduced to a variety of computer vision tasks, having demonstrated a more appealing performance than previous works. DETR (Carion et al., 2020), i.e. a typical successful transformer-based method is presented as the first end-to-end framework for object detection, achieving comparable performance with previous methods. Unfortunately, this is achieved at the expense of many more epochs to converge, which is quite slow. Many works are dedicated to resolving the convergence speed issue, e.g., (Zhu et al., 2021) propose deformable attention to speed up the training. In recent years, DETR has fostered similar research fields, e.g. extending DETR-like models to the RVOS. MTTR (Adam et al., 2022) models this task as a sequence prediction via a multi-modal transformer. Following this work, ReferFormer (Wu et al., 2022b) replaces the transformer used in MTTR with the deformable transformer (Zhu et al., 2021), achieving much faster training speed. Nevertheless, ReferFormer still has unfavorable limitations of lengthy training progress. In this work, our method is built on top of ReferFormer, and multi-referring features aggregation is proposed, enabling fewer epochs to converge and higher segmentation accuracy.

### 3 METHOD

In this section, we first review the model architecture for referring video object segmentation that the proposed method is built upon. Then, the proposed Neural Expression Generation module which is the core component to model multiple referring expressions is introduced, followed by the optimization of sampling strategy and losses. Figure 2 depicts the model architecture of the proposed method.

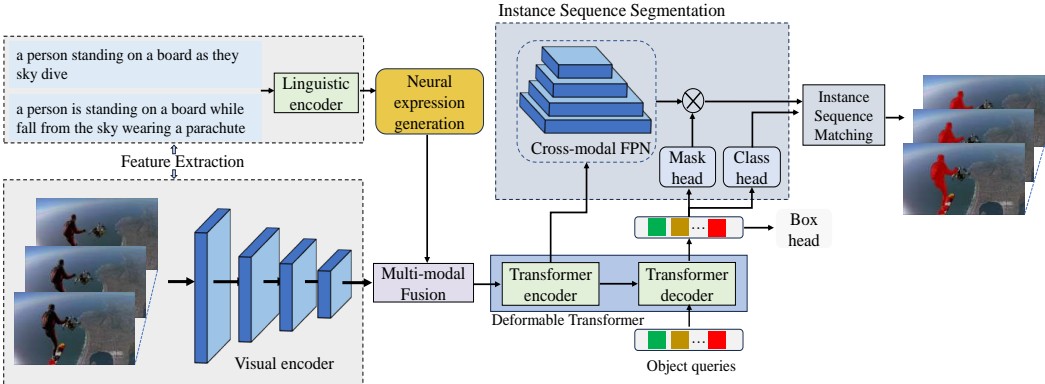

Figure 2: The model architecture of the proposed method.

### 3.1 PRELIMINARIES

**Feature Extraction.** The model begins with the visual encoder and the linguistic encoder for visual feature extraction and text feature extraction respectively. The backbone of the visual encoder takes

as input a video clip with $T$ frames of the resolution $H \times W$, denoted as $V_T \in R^{T \times 3 \times H \times W}$, and outputs multi-scale pixel-level features $F_T^l \in R^{T \times C^l \times H^l \times W^l}$, where $l$ denotes the spatial scale and $C^l$ denotes the feature channels at the scale $l$. Regarding the linguistic encoder, it takes as input the referring expression. The word features are extracted by the weights-sharing linguistic encoder, denoted as $L$.

**Multi-modal Fusion.** After the projection of multi-scale visual features to the same channel $C$, the language features cross-attend to the visual features in a multi-scale manner, outputting multi-scale fused features $T \times C \times H^l \times W^l$.

**Deformable Transformer.** Considering the computational burden and memory footprint, the temporal dimension of fused features is moved to the batch dimension beforehand. The encoder, composed of a stack of deformable self-attention layers, progressively attend to the early fused features at each spatial scale, so as to capture the global context of each frame. The query embedding associated with a learnable positional encoding is input to the decoder comprising of a stack of deformable attention layers to query the instance features from encoded memories. Note that the query embedding is initialized by text features from the linguistic encoder instead of zero-values, leading to considerable performance gains. This decoder outputs $N_q$ instances per frame.

**Instance Sequence Segmentation.** The Cross-Modal Feature Pyramid Network (CM-FPN) is employed to upsample multi-level low-resolution encoded memory features to generate high-resolution features $F_{seg}$. Meanwhile, multiple prediction heads implemented by different Feed Forward Networks independently decode the instance embedding from the decoder into the class label and bounding box coordinates. Moreover, the instance embeddings are transformed into dynamic convolutional weights and then features $F_{seg}$ are passed through dynamic convolutional layers to obtain mask predictions $\hat{y}$.

**Instance Sequence Matching and Losses.** At the last step, the bipartite matching and loss calculation between the instance sequence predictions and the corresponding ground truth are sequentially performed. Specifically, let us denote by $\hat{y} = \{\hat{y}_i\}_{i=1}^{N_q}$ the set of $N_q$ predictions, and by $y$ the ground truth. Next, a permutation of $N_q$ elements ($i \in S_{N_q}$) are searched with the minimum pair-wise matching cost between the two sets using Eq. 1 and Eq. 2.

$$\hat{i} = \underset{i \in S_{N_q}}{\arg \min} L_{match}(y, \hat{y}_i) \tag{1}$$

$$\begin{aligned} L_{match}(y, \hat{y}_i) = \lambda_{cls} L_{cls}(y, \hat{y}_i) + \lambda_{box} L_{box}(y, \hat{y}_i) \\ + \lambda_{mask} L_{mask}(y, \hat{y}_i) \end{aligned} \tag{2}$$

where $\lambda_{cls}$, $\lambda_{box}$, $\lambda_{mask} \in R$ are hyper-parameters, and $L_{cls}$, $L_{box}$, mask loss $L_{mask}$ represents classification loss, bounding box loss and mask loss respectively. Finally, the total loss $L_{match}$ for all pairs matched in the previous step is optimized for training the whole network.

## 3.2 NEURAL EXPRESSION GENERATION

Instead of feeding one referring expression to a deep neural network, we draw inspiration from an early work (Bellver et al., 2023) (experimental results from their various combinations of visual attributes illustrate the full-length referring expression improves performance, especially for non-trivial samples), and propose to introduce multiple referring expressions to learn the referring expression representation for improving the segmentation quality.

A straightforward approach to handling the referring expression variation is the concatenation of multiple linguistic features, dubbed as "MRE Concat". The concatenation is placed before early fusion, since this early fusion is surprisingly found much more effective compared to intermediate and late fusion. Specifically, multiple referring expressions, denoted by $RE = \{RE_i\}_{i=1}^N$, are fed into the Linguistic encoder to generate the corresponding linguistic features $L = \{L_i\}_{i=1}^N$ respectively. $N$ means the number of referring expressions. The concatenated fashion is shown in Eq. 3.

$$L_{Concat} = Concat(L_1, L_2, ..., L_N) \tag{3}$$

The concatenated features usually involve more attributes, being beneficial to the accuracy of separation of objects. However, this approach may not be sufficient to disregard useless features, causing distraction in mask outputs. Therefore, we propose to approximate the referring expression representation with a multi-layer perceptron (MLP) network and optimize the weights of the MLP network

to map MRE features to the more representative RE features.

$$L_{MLP} = MLP(L_{Concat}) \tag{4}$$

To make the MLP well-trained, we follow a common practice (Vaswani et al., 2017; Carion et al., 2020), i.e. to use residual connections, dropout and layer normalization, cf. Eq. 5.

$$L_{NEG} = L_q + Norm(Dropout(L_{MLP}))) \tag{5}$$

### 3.3 Optimization improvements

**Sampling points in Deformable Attention.** To quickly make attention weights $A_{mqk}$ focus on instance features from the encoded memory, the deformable attention is used in the pre-training phase, which attends to only a sparse set of points around the object of interest instead of all possible points. The multi-head attention features are computed by Eq. 6,

$$DeformAttn(z_q, P_q, x) = \sum_{j=1}^{J} W_j [\sum_{k=1}^{K} A_{jqk} \cdot W_j' x(P_q + P_{jqk})]) \tag{6}$$

where $k$ denotes the number of sampling points and $j$ is the index of attention head. $K$ is the number of total sampled points. $J$ is the number of total attention heads. $W_j'$ and $W_j$ are learnable weights.

After pre-training, the attention weights could be more or less focused on instance features in the new data. Thus, the querying instance in a global context seems more potential to boost performance. In order to aggregate more context to segment referred dynamic objects, in the fine-tuning phase, more points with larger $K$ are sampled around the object instances. Table 2 (b) shows that the improvement is achieved by optimizing the sampling of the deformable attention.

**Instance Sequence Matching and Losses.** To accurately sample the points around the center of reference points, the training begins with the supervision of a bounding box. Once the object instances are localized, we find the movement of bounding box supervision improves the segmentation accuracy. Therefore, the bounding box supervision is not used during the fine-tuning.

### 3.4 Implementation Details

The visual encoder in the proposed method is implemented as different visual backbones, including ResNet (He et al., 2016), swin transformer (Liu et al., 2021) and video swin transformer (Liu et al., 2022), as similarly done in ReferFormer. Our method is pre-trained on MS-COCO (Lin et al., 2014) dataset with referring expressions, i.e. Ref-COCO/+ (Yu et al., 2016) and Ref-COCOg (Mao et al., 2016), then fine-tuned on the Ref-Youtube-VOS (Seo et al., 2020) training set. Regarding the fine-tuning phase, we adopt the almost same hyperparameters with ReferFormer except that the total batch size is 8, the maximum image size is 448, and the initial learning rate is 5e-5. The whole fine-tuning took about 5 hours on 4 Nvidia RTX A6000 GPUs. No post-processing techniques are used in the inference phase.

## 4 Experiments

### 4.1 Datasets and Evaluation Metrics

**Ref-Youtube-VOS** dataset (Seo et al., 2020) is a large-scale RVOS dataset, which covers 3,978 YouTube videos from the YouTube-VOS dataset (Xu et al., 2018) with around 15k language expressions. 6,388 unique objects are included in the train set with 12,913 expressions and 1,063 unique objects are included in the validation set with 2,096 expressions. On average, each object has around 2 referring expressions and each expression has more or less 10 words. Since the annotations of the validation set are not publicly available, our validation results are reported by submitting mask predictions to the evaluation server.

**Ref-DAVIS17** dataset (Khoreva et al., 2018) has 1544 expressions for 205 objects in 90 videos from DAVIS17 dataset (Pont-Tuset et al., 2017). This dataset is split into 60 videos and 30 videos for training and validation, respectively. The results of the validation set are reported as done in the compared methods.

**A2Dre+** dataset (Bellver et al., 2023) is built upon the A2D-Sentence (Gavrilyuk et al., 2018) test

set that collects language expressions for 746 videos from the A2D dataset (Xu et al., 2015). This dataset is extended from the A2Dre dataset (Bellver et al., 2023) that contains 433 non-trivial expressions (corresponding to the referred object that is not the only object of a certain object class in a video) for 190 videos selected from the A2D-Sentence test set. 4 major semantic categories out of the 7 semantic categories in A2Dre, i.e. appearance, spatial location, motion, and static, are selected. With the available semantic categories, each expression is augmented by either including the semantic category or removing the semantic category.

**Metrics.** The standard metrics are used: region similarity $\mathcal{J}$, contour accuracy $\mathcal{F}$, and average value of $\mathcal{J}$ and $\mathcal{F}$. All results on Ref-Youtube-VOS are evaluated using the online validation server and results on Ref-DAVIS17 are evaluated by the official code. Following the previous work (Wu et al., 2022b), Overall IoU and Mean IoU are used to evaluate the RVOS models on the A2Dre+ dataset, which computes the ratio between the total intersection and the total union area over the test samples, and calculates the averaged IoU over all the test samples respectively.

## 4.2 Ablation Experiments

All ablation experiments are conducted on the Ref-Youtube-VOS validation set, and the Video-Swin-T is used as the backbone of the visual encoder.

**Impact of removing bounding box supervision.** Bounding box loss summing up the L1 loss and GIoU loss is used for object location supervision. Here we verify if removing bounding box loss helps the improvement of the accuracy. Specifically, we conducted several fine-tuning experiments with various combinations of coefficients for bounding box losses between 0 (i.e. without) and 2. As is shown in Table 2 (a), we could learn that the higher score is achieved by weakening the bounding box effect during fine-tuning, and the best accuracy is obtained when this loss is completely removed, probably because the global context information helps to accurately segment the referred object.

**Impact of number of sampling points.** We discuss the impact of the number of sampling points $K$ in deformable attention. As shown in Table 2 (b), the increase from the original $K = 4$ to $K = 16$ witnesses the improvement of $\mathcal{J}$ & $\mathcal{F}$ from 60.8 to 61.1, illustrating that the performance is possiblely to be boosted from a larger context of the encoded memory.

**Impact of NEG module.** Table 2 (c) shows various design choices for multi-referring feature aggregation. The "Single RE" listed in this Table is regarded as the baseline. Compared to the Single RE, the proposed NEG achieves gains of 1.6% in $\mathcal{J}$ & $\mathcal{F}$, 1.7% in $\mathcal{J}$, and 1.6% in $\mathcal{F}$ respectively. "MRE Ranking" ranks the likelihood of candidate indexes from MRE in the inference phase. Compared to the MRE Ranking, the number of epochs that the MRE Concat and NEG takes is $2\times$ less.

| L1 | GIoU | $\mathcal{J}$ & $\mathcal{F}$ | $\mathcal{J}$ | $\mathcal{F}$ |
|----|------|------|------|------|
| 2 | 2 | 59.6 | 57.9 | 61.4 |
| 1 | 1 | 60.4 | 58.7 | 62.0 |
| 0.5 | 0.5 | 60.4 | 58.8 | 62.0 |
| 0.25 | 0.25 | 60.5 | 58.8 | 62.2 |
| 0 | 0 | **60.8** | **59.1** | **62.4** |

(a): impact of with and without bounding box loss.

| | $\mathcal{J}$ & $\mathcal{F}$ | $\mathcal{J}$ | $\mathcal{F}$ |
|----|------|------|------|
| $K$=4 | 60.8 | 59.1 | 62.4 |
| $K$=16 | **61.1** | **59.5** | **62.7** |

(b): impact of $K$ used in deformable attention.

| | $\mathcal{J}$ & $\mathcal{F}$ | $\mathcal{J}$ | $\mathcal{F}$ | #Eps |
|----|------|------|------|------|
| Single RE | 61.1 | 59.5 | 62.7 | 6 |
| MRE Ranking | 62.6 | 61.1 | 64.2 | 6 |
| MRE Concat | 62.2 | 60.4 | 64.1 | **3** |
| NEG | **62.7** | **61.2** | **64.3** | **3** |

(c): impact of various designs for the MRE.

Table 2: Ablation studies of model settings and model design choices on the Ref-Youtube-VOS validation set. #Eps indicates the number of fine-tuning epochs.

## 4.3 Performance on Ref-Youtube-VOS and Ref-DAVIS17

**Quantitative comparisons.** Table 3 demonstrates the quantitative comparisons between the proposed method and the state-of-the-art methods (URVOS (Seo et al., 2020), YOFO (Li et al., 2022), LOCATER (Liang et al., 2023), CSTM (Hui et al., 2023), LBDT (Ding et al., 2022), MTTR (Adam et al., 2022), ReferFormer (Wu et al., 2022b)). Here, for the sake of fair comparisons, the compared methods using the same backbone are put together since the backbone makes a big difference in the

Table 3: Performance comparison results on the Ref-Youtube-VOS validation set and Ref-DAVIS17 validation set. NT indicates the number of epochs pre-trained on Ref-COCO/+ (Yu et al., 2016), Ref-COCOg (Mao et al., 2016), and NF is the number of epochs fine-tuned or trained on Ref-Youtube-VOS training set Seo et al. (2020).

| Method | backbone | Epochs (NT+NF) | Ref-Youtube-VOS | | | Ref-DAVIS17 | | |
|--------|----------|----------------|-----------------|---|---|-------------|---|---|
| | | | $\mathcal{J}\&\mathcal{F}$ | $\mathcal{J}$ | $\mathcal{F}$ | $\mathcal{J}\&\mathcal{F}$ | $\mathcal{J}$ | $\mathcal{F}$ |
| URVOS | ResNet-50 | 0+120 | 47.2 | 45.3 | 49.2 | 51.5 | 47.3 | 56.0 |
| YOFO | ResNet-50 | 0+150 | 48.6 | 47.5 | 49.7 | 53.3 | 48.8 | 57.8 |
| LOCATER | ResNet-50 | 0+30 | 50.0 | 48.8 | 51.1 | - | - | - |
| CSTM | ResNet-50 | 0+15 | 49.3 | 48.2 | 50.5 | 54.5 | - | - |
| LBDT-4 | ResNet-50 | 0+15 | 49.4 | 48.1 | 50.6 | 54.5 | - | - |
| ReferFormer | ResNet-50 | 12+6 | 55.6 | 54.8 | 56.5 | 58.5 | 55.8 | 61.3 |
| Ours | ResNet-50 | 12+3 | **58.0** | **57.0** | **59.1** | **59.4** | **56.6** | **62.2** |
| ReferFormer | Swin-T | 12+6 | 58.7 | 57.6 | 59.9 | 55.8 | 53.2 | 58.3 |
| Ours | Swin-T | 12+3 | **61.7** | **60.3** | **63.1** | **59.7** | **56.8** | **62.5** |
| ReferFormer | Swin-L | 12+6 | 62.4 | 60.8 | 64.0 | 60.5 | 57.6 | 63.4 |
| Ours | Swin-L | 12+3 | **65.4** | **63.5** | **67.2** | **62.6** | **59.9** | **65.3** |
| MTTR | Video-Swin-T | 0+30 | 55.3 | 54.0 | 56.6 | - | - | - |
| ReferFormer | Video-Swin-T | 12+6 | 59.4 | 58.0 | 60.9 | 59.7 | 56.6 | 62.8 |
| Ours | Video-Swin-T | 12+3 | **62.7** | **61.2** | **64.3** | **61.0** | **58.0** | **64.0** |

RVOS task. Note that all variants of our method share the same parameters except the backbones. In this comparison, the proposed method surpasses the previous works in terms of segmentation accuracy by a significant margin. Besides, our method consistently outperforms ReferFormer by more than 2.4% $\mathcal{J}\&\mathcal{F}$ on the Ref-Youtube-VOS dataset irrespective of the backbones. Likewise, superior segmentation quality is achieved on the Ref-DAVIS17 dataset. Apart from the segmentation quality metric, our method also shows superior performance in terms of training speed, taking the fewest epochs to converge among all the top-performing methods. Reaching the same quality of segmentations, our method seems around 4× faster than the ReferFormer in terms of fine-tuning convergence speed, cf. Figure 4 in Appendix A.1.

**Qualitative comparisons.** Figure 3 shows the visual comparison results between our method and ReferFormer. As seen in Figure 3, our method significantly improves the segmentation quality, especially for non-trivial samples (i.e. multiple objects of the same class), while ReferFormer sometimes might misidentify one object as another object, miss some keywords, or not fully understand the sentence. Besides, the order of the two referring expressions makes little difference in our results.

## 4.4 PERFORMANCE ON A2DRE+

The A2Dre+ is further used to verify the effectiveness of multi-referring feature aggregation for the RVOS. The Video-Swin-T is used as the backbone of the visual encoder in all compared methods.

**Quantitative comparisons.** From Table 4, we observe that, compared with only feeding the referring expression w/o APP (without appearance category) or only feeding the referring expression w/o SP (without spatial location category), ReferFormer is more beneficial from feeding the concatenation of referring expressions w/o APP and w/o SP (i.e. 57.4% Overall IOU (w/o APP + w/o SP) higher than 52.2% (w/o APP) and 50.0 (w/o SP)). With the same referring expressions (w/o APP + w/o SP), Ours gets a much higher 60.2% Overall IOU than 57.4% of ReferFormer, which illustrates the effectiveness of the proposed Neural expression generation module for multi-referring feature aggregation.

## 5 CONCLUSION

In this paper, we present a novel network fed with multiple referring expressions for referring video object segmentation. A novel neural expression generation module is proposed to obtain complete

$RE_1$: the first giraffe from the right
$RE_2$: the giraffe is on the right side of the railing and has its head hanging over and eating grass

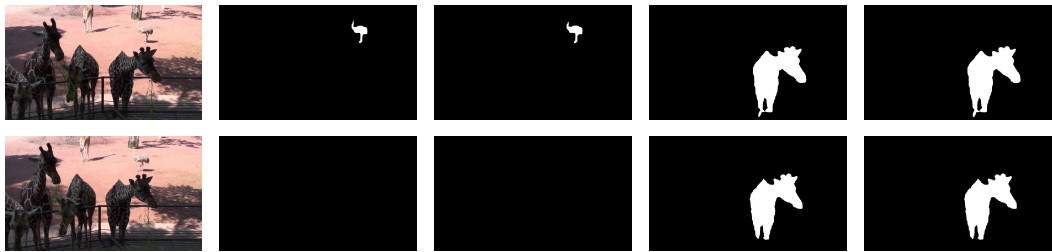

$RE_1$: a lady strapped to a man both skydiving
$RE_2$: a person with red glasses is in front of another and both are going to jump out of the plane

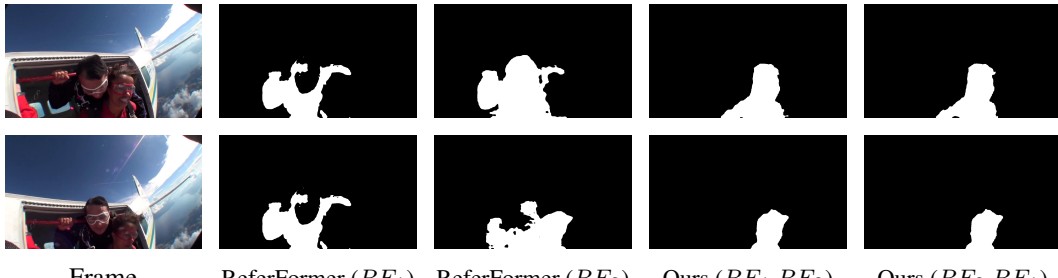

| Frame | ReferFormer ($RE_1$) | ReferFormer ($RE_2$) | Ours ($RE_1$,$RE_2$) | Ours ($RE_2$,$RE_1$) |

Figure 3: Visual comparison results on the Ref-Youtube-VOS validation set.

Table 4: Performance comparison results on A2Dre+ dataset. Different kinds of referring expressions, annotated with semantic categories are included or not included, are used. APP: appearance category, SP: spatial location category.

| Referring Expression | Method | IoU | |
|---|---|---|---|
| | | Overall | Mean |
| w/o APP | ReferFormer | 52.2 | 50.3 |
| w/o SP | ReferFormer | 50.0 | 50.6 |
| APP+w/o SP | ReferFormer | 55.8 | 54.9 |
| APP+w/o SP | Ours | **59.4** | **55.2** |
| w/o APP+SP | ReferFormer | 56.1 | 54.0 |
| w/o APP+SP | Ours | **59.6** | **56.1** |
| w/o APP+ w/o SP | ReferFormer | 57.4 | 54.8 |
| w/o APP+ w/o SP | Ours | **60.2** | **56.8** |

and concise linguistic features by multi-referring feature aggregation. Moreover, optimization strategies are proposed to boost the model performance by different sampling strategies in cross-modal attention and different bounding box supervision for pre-training and fine-tuning. Evaluations are made on the two popular RVOS datasets, and experimental results show a superior performance than previous methods. We believe this work makes progress toward the modeling of referring expressions in segmenting the referred object, and hope it will open up the possibilities in exploring the essential understanding of referring expressions in referring video object segmentation.

## 6    REPRODUCIBILITY STATEMENT

The experimental results in the paper are reproducible with additional implementation details, which could be found in Appendix A.5.

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

# A APPENDIX

## A.1 COMPARISON OF THE ACCURACY AND THE FINE-TUNING SPEED

Figure 4 shows comparisons of the accuracy between ReferFormer (Wu et al., 2022b) and the proposed method under various fine-tuning epochs. The experimental results notably show the benefit of the proposed NEG module for multi-referring feature aggregation and optimization improvements on aspects of RVOS accuracy and fine-tuning convergence speed.

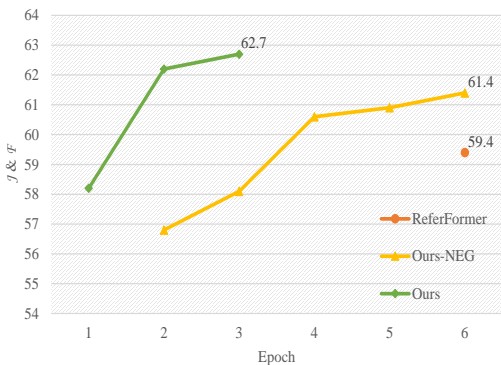

Figure 4: Performance comparison on the Ref-Youtube-VOS validation set (Seo et al., 2020). "Ours-NEG" means our method without the proposed NEG module. The higher $\mathcal{J}$ & $\mathcal{F}$ represents the better accuracy.

## A.2 INDIRECT IMPROVEMENT OF THE TRAINING SPEED

To further validate the efficiency of multi-referring feature aggregation on the training speed of the whole "pretrain-then-finetune" process, we also report the results from various epochs spent in the pre-train stage, as shown in Table 5.

Table 5: Performance comparison results on the Ref-Youtube-VOS validation set. NT indicates the number of epochs pre-trained on Ref-COCO/+ (Yu et al., 2016) and Ref-COCOg (Mao et al., 2016). NF is the number of epochs fine-tuned on the Ref-Youtube-VOS training set (Seo et al., 2020). †: 32 Nvidia V100 GPUs with 32GB memory used in ReferFormer (Wu et al., 2022b). Other results are obtained using 4 Nvidia RTX A6000 GPUs with 48GB memory.

| Method | Epochs (NT+NF) | Ref-Youtube-VOS $\mathcal{J}$ & $\mathcal{F}$ | $\mathcal{J}$ | $\mathcal{F}$ |
|---|---|---|---|---|
| ReferFormer | $12^{\dagger}+6^{\dagger}$ | 59.4 | 58.0 | 60.9 |
| Ours | $12^{\dagger}+3$ | **62.7** | **61.2** | **64.3** |
| Ours | 12+3 | 62.0 | 60.3 | 63.8 |
| Ours | 7+3 | 62.0 | 60.4 | 63.6 |
| Ours | 4+3 | 62.0 | 60.6 | 63.4 |

Comparing the second row (32 Nvidia V100 GPUs with 32GB memory in use) with the third row (4 Nvidia RTX A6000 GPUs with 48GB memory in use), our proposed model is still able to achieve high performance with limited GPU memory footprints. Comparing the first row with the last row, our method, spending only 1/3 epochs in the pre-training stage, still outperforms the ReferFormer (Wu et al., 2022b) by a significant margin, which demonstrates that the proposed method with the interaction of multiple referring expressions is capable of improving the speed of the whole training.

## A.3 MORE COMPARISONS WITH RECENTLY PUBLISHED METHODS

Table 6 shows comparisons with recently published methods, including $R^2$-VOS (Li et al., 2023), OnlineRefer (Wu et al., 2023), HTML (Han et al., 2023). The results of all compared methods are obtained from the published paper. The proposed method still outperforms these recently published methods by a non-negligible margin with the fewest training epochs.

Table 6: Performance comparison results on the Ref-Youtube-VOS validation set. NT indicates the number of epochs pre-trained on Ref-COCO/+/g datasets and NF is the number of epochs fine-tuned on Ref-Youtube-VOS training set (Seo et al., 2020).

| Method | backbone | Epochs (NT+NF) | Ref-Youtube-VOS $\mathcal{J}$ & $\mathcal{F}$ | $\mathcal{J}$ | $\mathcal{F}$ |
|---|---|---|---|---|---|
| ReferFormer | ResNet-50 | 12+6 | 55.6 | 54.8 | 56.5 |
| $R^2$-VOS | ResNet-50 | 12+6 | 57.3 | 56.1 | 58.4 |
| OnlineRefer | ResNet-50 | 12+6 | 57.3 | 55.6 | 58.9 |
| HTML | ResNet-50 | 12+6 | 57.8 | 56.5 | 59.0 |
| Ours | ResNet-50 | **12+3** | **58.0** | **57.0** | **59.1** |
| ReferFormer | Swin-T | 12+6 | 58.7 | 57.6 | 59.9 |
| $R^2$-VOS | Swin-T | 12+6 | 60.2 | 58.9 | 61.5 |
| Ours | Swin-T | **12+3** | **61.7** | **60.3** | **63.1** |
| OnlineRefer | Swin-L | 12+6 | 63.5 | 61.6 | 65.5 |
| HTML | Swin-L | 12+6 | 63.4 | 61.5 | 65.3 |
| Ours | Swin-L | **12+3** | **65.4** | **63.5** | **67.2** |
| $R^2$-VOS | Video-Swin-T | 12+6 | 61.3 | 59.6 | 63.1 |
| HTML | Video-Swin-T | 12+6 | 61.2 | 59.5 | 63.0 |
| Ours | Video-Swin-T | **12+3** | **62.7** | **61.2** | **64.3** |
| OnlineRefer | Video-Swin-B | 12+6 | 62.9 | 61.0 | 64.7 |
| HTML | Video-Swin-B | 12+6 | 63.4 | 61.5 | 65.2 |
| Ours | Video-Swin-B | **12+3** | **65.3** | **63.6** | **66.9** |

## A.4 VISUALIZATION

### A.4.1 RESULTS ON REF-YOUTUBE-VOS

In Figure 5, we visualize a sequence of mask predictions of the proposed method. These examples show that our method is capable of predicting meaningful and accurate segmentation masks associated with the input referring expressions. Additionally, our method is able to produce temporarily consistent segmentation masks.

### A.4.2 RESULTS ON A2DRE+

Figure 6 shows the visual predictions of the proposed method using the fine-tuned model on Ref-Youtube-VOS, in which textual inputs are composed of different combinations of semantic categories. Without any fine-tuning on this dataset, our proposed method is capable of identifying the referred object, and segmenting the object in a high quality.

## A.5 ADDITIONAL IMPLEMENTATION DETAILS

Our model is built upon the ReferFormer ((Wu et al., 2022b)) model, which is publicly available on the github "ReferFormer" repository. Regarding the pre-training stage, we adopt the same hyperparameters with ReferFormer except that the total batch size is 16. Regarding the results on Ref-DAVIS17 dataset and A2Dre+ dataset, they are both predicted by the fine-tuned model on Ref-Youtube-VOS dataset.

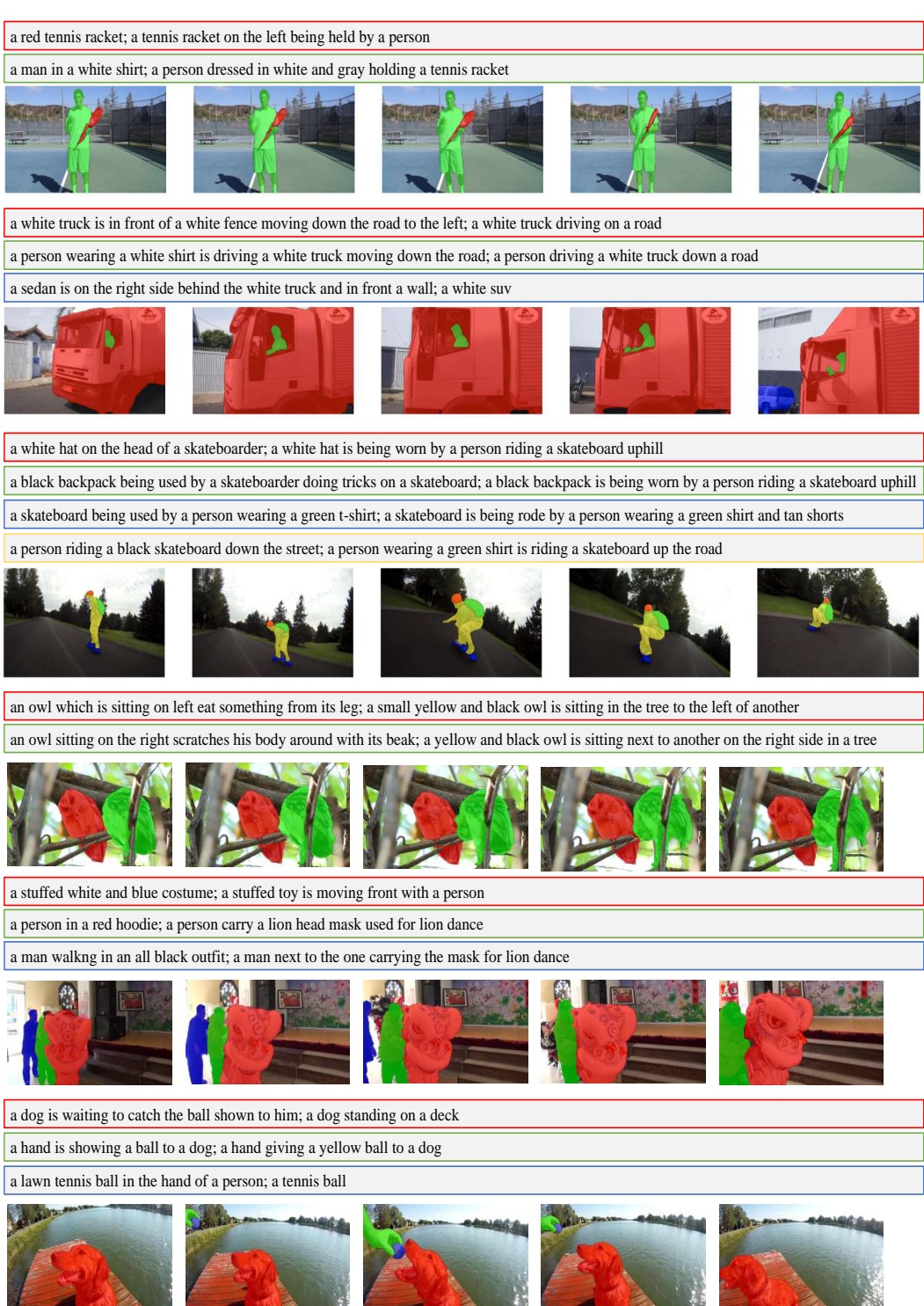

Figure 5: Visualization results on Ref-Youtube-VOS validation set. The color of the rectangle contour corresponds to the color of the mask overlaid on the referred object.

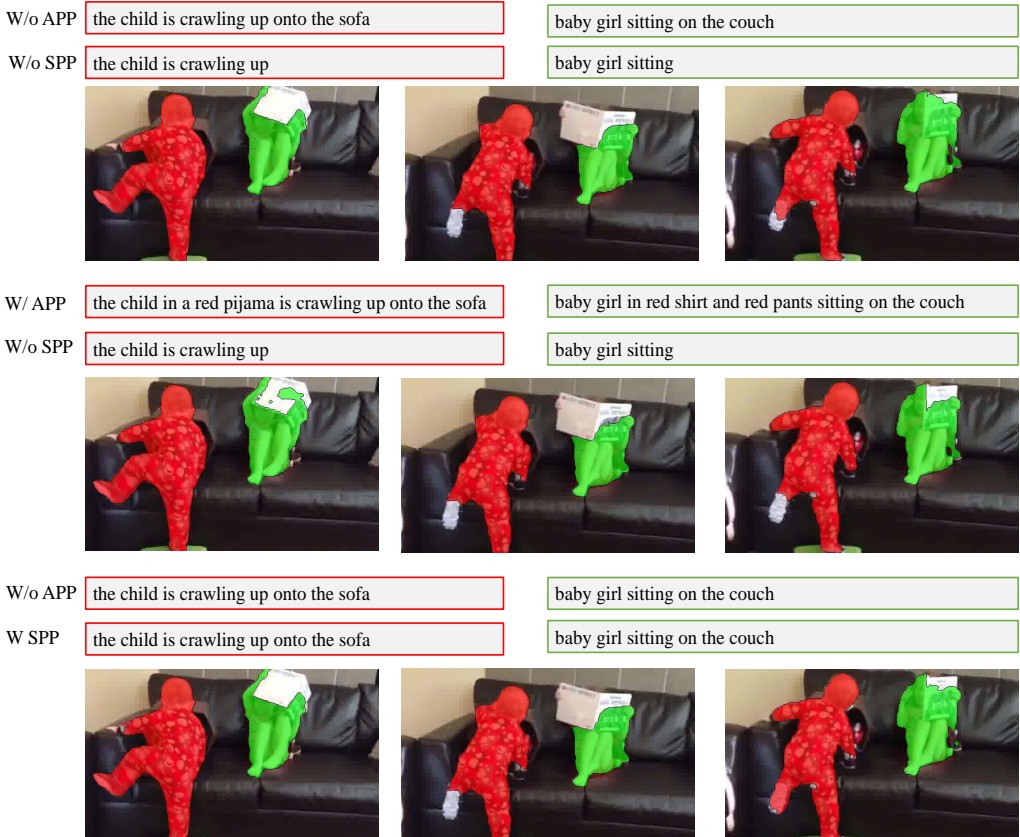

Figure 6: Visualization results on A2Dre+ dataset. The color of the rectangle contour corresponds to the color of the mask overlaid on the referred object.

