# OpenReview forum: "Referring Expression Matters: Multi-referring Feature Aggregation for Referring Video Object Segmentation"
_ICLR.cc/2024/Conference — ICLR 2024 Conference Withdrawn Submission_

### Official Review · Reviewer_QT6n · 2023-10-30

**Soundness:** 2 fair
**Presentation:** 3 good
**Contribution:** 2 fair
**Rating:** 5
**Confidence:** 4

**Summary:**

The paper focuses the task of Referring Video Object Segmentation and introduces to integrate multiple referring expressions to boost performance. A neural expression generation module is proposed to create complementary features from these expressions, which not only improves object identification accuracy but also accelerates training convergence. Experimental results on popular RVOS datasets are presented.

**Strengths:**

(1)	The paper explores the effect of multiple referring expressions for RVOS, which is interesting.

(2)	This paper is well-written and easy to follow.

**Weaknesses:**

(1) Although the authors present an interesting motivation, suggesting that adjusting referring expressions could enhance segmentation performance, the method proposed does not fully align with this motivation. The reviewer, after going through the introduction, expect to find how the unclear parts within referring expressions are identified and improved. However, the authors merely concatenate multiple referring expressions.

(2) The paper's contribution mainly involves adding an MLP to ReferFormer to merge multiple referring expressions. However, this incremental addition lacks further in-depth consideration, i.e., what kind of scenarios need multiple inputs, how the extent of overlap and divergence between referring expressions affects final performance. Consequently, the contribution of the paper is limited.

(3) The experimental comparisons are unfair. While the proposed method uses multiple referring expressions as input, the compared methods utilize only one expression. To truly demonstrate the impact of the integration of MRE, a more comprehensive comparison should involve merging results from different expressions in other methods. This would effectively showcase the performance gains derived from exploring relationships within referring expressions.

**Questions:**

See weaknesses.

---

### Official Review · Reviewer_SAH4 · 2023-10-30

**Soundness:** 1 poor
**Presentation:** 1 poor
**Contribution:** 1 poor
**Rating:** 1
**Confidence:** 3

**Summary:**

A Referring Video Object Segmentation method is proposed. However, the motivation is not clear. The details of most of the methods are not explained.

**Strengths:**

The picture of the model architecture of the proposed method is clear.

**Weaknesses:**

The writing is too bad. The details of the Multi-modal Fusion are not explained clearly. No referenced paper is mentioned in Deformable Transformer and Instance Sequence Segmentation. It's quite hard to understand the paper.

**Questions:**

How to do Multi-modal Fusion? What's the structure of the Deformable Transformer? What is Cross-Modal Feature Pyramid Network (CM-FPN) ?

---

### Official Review · Reviewer_s2MZ · 2023-11-01

**Soundness:** 2 fair
**Presentation:** 2 fair
**Contribution:** 2 fair
**Rating:** 3
**Confidence:** 2

**Summary:**

This paper proposes a referring video object segmentation method via a multi-referring feature aggregation mechanism. This mechanism can effectively obtain complementary features with less redundancy, which is not only helpful in identifying the referred object, but also speeds up the training convergence. Experimental results show the effectiveness and superiority of the proposed method.

**Strengths:**

+ The multiple referring expressions can generate a complete and concise linguistic feature, experimental results also show the effectiveness of the proposed strategy.
+ The proposed method can achieve better training convergence.
+ The proposed method achieves the new SOTA and outperforms the second-best by a large margin

**Weaknesses:**

- The novelty of the proposed method is somewhat limited. The main contribution is the neural expression generation via multiple-referring expressions. It seems that this aggregation strategy is simple and lacks insights.
- The authors declare that they proposed different sampling strategies in cross-modal attention for pre-training and fine-tuning to boost the model performance. However, the illustration of this sampling strategy is unclear, and the differences with existing sampling strategies are also unclear. Also, there are no experimental results to support this assertion.
- In Eq.3, the authors used the concat operation but in Table 2(c), the proposed NEG is different MRE Cocat, so the reason is unclear.
- The authors do not show the training convergence in the pre-training strategy, So, it is hard to assert the proposed method achieves faster convergence only by verifying it in the fine-tuning stage.
 - I think the comparison is somewhat unfair. The batch size is different. It mainly influences the training convergence and even the performance.

**Questions:**

Please seeing the weaknesses.